# Characterization Studies on Graphene-Aluminium Nano Composites for Aerospace Launch Vehicle External Fuel Tank Structural Application

**DOI:** 10.3390/ma15175907

**Published:** 2022-08-26

**Authors:** Joel Jayaseelan, Ashwath Pazhani, Anthony Xavior Michael, Jeyapandiarajan Paulchamy, Andre Batako, Prashantha Kumar Hosamane Guruswamy

**Affiliations:** 1School of Mechanical Engineering, Vellore Institute of Technology, Vellore 632014, India; 2School of Mechanical Engineering, Coventry University, Priory St, Coventry CV1 5FB, UK; 3General Engineering Research Institute, Liverpool Jhon Moores University, Merseyside L3 5UX, UK; 4Indian Institute of Technology, Chennai 600036, India

**Keywords:** Aluminium-Graphene composites, metallurgical characterisation, launch vehicle external fuel tank structure, nano composites

## Abstract

From the aspect of exploring the alternative lightweight composite material for the aerospace launch vehicle external fuel tank structural components, the current research work studies three different grades of Aluminium alloy reinforced with varying graphene weight percentages that are processed through powder metallurgy (P/M) route. The prepared green compacts composite ingots are subjected to microwave processing (Sintering), hot extruded, and solution treated (T6). The developed Nano-graphene reinforced composite is studied further for the strength–microstructural integrity. The nature of the graphene reinforcement and its chemical existence within the composite is further studied, and it is found that hot extruded solution treated (HEST) composite exhibited low levels of carbide (Al_4_C_3_) formations, as composites processed by microwaves. Further, the samples of different grades reinforced with varying graphene percentages are subjected to mechanical characterisation tests such as the tensile test and hardness. It is found that 2 wt% graphene reinforced composites exhibited enhanced yield strength and ultimate tensile strength. Microstructural studies and fracture morphology are studied, and it is proven that composite processed via the microwave method has exhibited good ductile behaviour and promising failure mechanisms at higher load levels.

## 1. Introduction

Graphene has been the focus material in the current trend, owing to its exceptional physical and mechanical properties. Graphene comprises carbon atoms arranged in a honeycomb structure in a single layer, and this monolayer surface is 2D in nature [1]. It was realized that graphene is the source of building blocks for fullerenes (0D), nanotubes (1D), and can be arranged into a number of sheets, forming 3D graphite [2]. This material has been widely researched in the past twenty years as possible reinforcement material in polymer and metal matrix composites. Graphene is generally produced with various processing techniques, such as exfoliation and cleavage [3], Epitaxial growth [4], chemical vapor deposition, and chemically derived graphene [5]. Graphene is the building block for carbon nanotubes; unzipping CNT [6] can give rise to graphene as well. Characterisation of graphene is a challenging task, as chemical and compositional analysis such as X-ray Diffraction (XRD), X-ray Florescence Spectroscopy (XRF), and other spectroscopy methods reveal graphene as carbon [7]. However, methods such as Fourier transform infrared (FTIR) and Transmission Electron Microscope (TEM) analysis help in studying the reinforced graphene [8]. In Aerospace External fuel tank structural applications that demand an ‘enhanced strength to weight ratio’ as important criteria, graphene-based Aluminium composite can suffice. Graphene reinforced metal matrix composites possess enhanced properties compared to the monolithic alloy by exhibiting bio-inspired lamellar fine grain microstructure, which is good for the launch vehicle external fuel tank structural application [9]. Aluminium metal matrix composites reinforced with graphene could be a potential material to serve in various aerospace applications with specific foci in the launch vehicle external fuel tank structural applications and various other space exploration strategic industries. The present research focuses on fabrication and development of graphene-reinforced Aluminium metal matrix composites through microwave sintering via powder metallurgy route followed by the hot extrusion method (comparable to the rolled plates (Airware–AA 2195 Aluminium lithium used in external tank structural application)) and studying the influence of graphene addition by metallurgical and mechanical characterisations.

## 2. Materials and Methods

### 2.1. Materials

Aluminium (Matrix material), possessing an excellent strength-to-weight ratio, is a non-ferrous alloy used in various strategic applications such as space exploration, aircraft, and several structural areas, which are the reasons for considering potential material in this study. The research experiments were focused on three different grades, i.e., AA 2024 (Copper based alloys), AA 6061 (Magnesium and Silicon based alloy), and AA 7075 (Zinc based alloy) aluminium powders (gas atomized) in the order of 10 Microns average particle diameter and up to 99.98 wt% purity, which was procured from Ampal Inc. (Palmerton, PA, USA).

Graphene as a potential reinforcing material, with its exceptional mechanical and physical properties and an average of ~5–10 μm (X-Y dimensions) and specific surface area of 1200 to 1450 m^2^/g, was purchased from Angstron Materials Inc., (Dayton, OH, USA). The same has been used to reinforce the alloys AA2024, AA6061, and AA7075 matrices with varying weight composition percentage by 1 wt%, 1.5 wt%, 2 wt%, 2.5 wt%, and 3 wt%.

### 2.2. Microwave Processing and Hot Extrusion of Aluminium-Graphene Nano Composite

Aluminium graphene composite was initially fabricated by employing the stir-Squeeze casting method; however, the homogenous dispersion of graphene was observed as a challenge due to agglomeration and graphene being less dense; it flies off the liquid molten pool. Powder metallurgy was found to be a suitable method for fabrication of AA (2024, 6061, 7075)-Graphene nano composites.

The process involves processing and extraction of fine layers of graphene by ultra-sonication and consolidated with calculated weight of metal powders introduced which is stirred for a period of 60 min to obtain uniform slurry in the shear mixer and then dried in a vacuum for 2 h at 60 °C. To ensure homogeneous distribution, the mixture is ball milled for 30 min with a ball-to-powder ratio of 10:1. Tungsten carbide balls of 10 mm were utilised. 

The homogenised composite mixture with different combinations of graphene percentage (1 wt%, 1.5 wt%, 2 wt%, 2.5 wt%, and 3 wt%) was further mixed with zinc stearate binders and polyvinyl alcohol (PVA) as functionaliser, and compaction was carried out with steel dies at a dwell time of 3 min at 500 MPa pressure to obtain Ø50 mm × 100 mm-long green compacts. Powder metallurgy involves various processes, which include weighing, blending, and consolidation of the powder mixture (Aluminium alloys with Graphene), compaction, and finally sintering (microwave). The green compacts are sintered in a microwave furnace for C for 15 min holding time in the continuous presence of an argon inert gas environment. The process parameters of sintering were arrived after a series of pilot studies and optimisation. Thus, sintered green billets are then subjected to hot extrusion to eliminate porosities and improve the density of composites (comparable to the hot rolled Aluminium Lithium alloy plates used in super lightweight (SLWT) External tank Structural application), and to obtain a definite shape of flat and 450 °round bars, as shown in Figure 1a–d, which is then solution treated and utilised to characterise its metallurgical, mechanical, and machinability studies.

## 3. Results

### 3.1. Morphology of as Received Graphene (Finite Element–Scanning Electron Microscope FE-SEM)

Figure 2 shows FE-SEM images of graphene Nano-flakes at 18 kx magnification obtained from carl-zeiss, Evo 18 model; it can be observed that graphene has a quasi-two-dimensional flaky structure. It is also noted that graphene has a huge surface area which plays a key role in wettability of graphene Nano flakes as reinforcements to matrix material. The surface is attributed to its planar structure in the X-Y plane with a Specific Surface Area (SSA) of 1200 to 1450 m^2^/g; the large surface area of graphene provides high wettability and contact area. The layer of graphene is stacked one over the other into agglomerations. However, it is necessary to ultra-sonicate the graphene during fabrication in order to prevent agglomeration of graphene sheets. 

### 3.2. Morphology of as Received and Exfoliated Graphene (High Resolution Transmission Electron Microscope HR-TEM)

When graphene-reinforced composites are studied and validated for external fuel tank structural application, the morphology of graphene used as a reinforcement must be examined using HR-TEM, which is a powerful tool to analyse the morphological and structural characteristics of graphene from the micro- to nanoscale in the order of atomic resolution. Figure 3, the image at 2500× resolution, shows few monolayers of graphene sheets with wrinkles and folds [10], which is also visible at the higher magnification of 14,500×. In Figure 4, the imaging conditions are set to expose the monolayer carbon atoms in white colour, and stacked monolayers (bilayer) with black atom contrast. The folds observed in the graphene sheets will have adverse effect on the composite strength and stiffness properties, which are the most expected property aspects for a SLWT external fuel tank structure under cryogenic conditions. Recent research reports, however, have experimented and recorded different values of strength and stiffness, possibly starting from the characteristic and inevitable wrinkling nature of added graphene in direction that is out-of-plane of graphene sheet with monolayer nature [11]. These wrinkles can be potentially observed as a result of out-of-plane flexural phonons or static wrinkling phenomena, which can be originated by uneven stress concentrations experienced during the graphene production technique, and they are accountable for the degradation of the mechanical properties when reinforced to the fabricated composite material in the current form [12]. Additionally, an alternative probable cause of wrinkling in the graphene layer may be mainly due to the occurrences of point defects at a measurable short distance, represented in cases like observed in the Stone-Wales defects [11].

Figure 5 reveals a vague lattice structure of carbon atoms in atomic resolution around 80 kX magnification, the aberration-corrected monochromatic image exposes “quasi-two-dimensional hexagonal lattice”. The inset in Figure 5 shows a bright-field phase contrast profile which was measured along a linear line in apparently a unique signature of monolayer graphene [11]. The image is further processed with superimposed hexagonal pattern to validate the graphene honeycomb lattice shown in Figure 6.

The electron diffraction pattern is instrumental in determining the layer and lattice parameters of the material examined. Figure 7 shows an ED pattern obtained from observing graphene flakes; the inner ring of {10¯10} pattern recorded more intense spots than outer ring {11¯20}, confirming the monolayer graphene flake [12,13]. This phenomenon is in good correlation with SAED patterns as well, where in case of a bilayer graphene sheet examination, outer rings were observed to be more intense.

### 3.3. Morphology of as Received Aluminium Alloy Powders

Morphological characteristics such as particle shape, size, texture, and surface area can be analysed using FESEM. Figure 8a–c shows the FE-SEM photographs of aluminium powders depicting its morphology, size, and distribution. It is clear that aluminium powders are spherical [13] and rounded in the three-dimensional state. Spherical and rounded particles have good apparent density and flow rate and also have smaller angle of repose enabling a good flow rate and inter-particle bonding and friction. In the present study, aluminium powders of ultrafine size are employed in the order of average particle diameter in the range 10–25 µm. Particle size is based on the characteristic features, such as maximum dimensions, mass, volume surface area, and minimum diameter; in the case of spherical powders, particle size is characterised based on their diameter [12].

To understand the suitability of the fabricated composite for the SLWT External tank structural application, the final composite properties matching the existing material need to be studied from the materials aspect, which can be influenced by factors such as grain size and the reinforcement-matrix strength microstructural integrity of the final extruded composite. In this direction, P/M fabrication process on materials involves different distributions of the particle size (10–25 µm) to ensure enhanced uniform properties in the final part. Generally, secondary operations such as laser treatment, bulk modification including heat treatment, and aging play major roles in controlling the grain size growth and eventually in controlling the mechanical properties of the final component [14].

Particle shape and size plays a significant role on the component’s final properties. Resultant grain size analysis is the fundamental characteristic of powder particles that affects other parameters such as materials flow rate, apparent density, processability, compressibility, and formability. Obtaining ultra-fine grained microstructure improves the mechanical properties such as ultimate tensile, compressive and rupture strengths, elongation to failure, tensile and compressive young’s modulus, shear modulus, density, and at last the thermal characteristics such as thermal conductivity, specific heat capacity, and thermal expansion coefficient, which are in agreement with the properties highlighted, outlined, and published by Metallic Materials Properties Development and Standardization (MMPDS), Federal Aviation Administration (FAA) [15].

In the process of developing a new alternative composite material combination for the existing SLWT external tank structural application, final composite materials grain size is engineered by using matrix average particle size of (~10–25 µm) and reinforcement average particle size of (~10 nm) [16,17]. The Hall Petch equation emphasizes the same relation between the final grain size to the recorded yield stress that is represented in Equation (1), where the grain size reduction will always improve the strength of the material. Fine metal powders result in poor apparent density, low flow rates, and high sinterability, while coarse powders have better apparent density and a reasonable flow rate but reduced sinterability [16,17,18].
(1)$$\sigma_{y}=\sigma_{0}+\frac{k_{y}}{\sqrt{d}}$$

### 3.4. EDS Analysis of as Received Al and Graphene

As aimed for the alternative composite material for existing SLWT external tank structural components such as the liquid oxygen tank, intertank, and liquid hydrogen system, it is very important to study the elemental composition of the matrix and reinforcement used in fabricating the composite [14]. In the FE-SEM, elemental composition was performed as and when morphology was observed. Figure 9a–c shows the elementary composition of Aluminium 2024, 6061, 7075 powders and graphene as given. It is clearly evident that the alloying elements present in the material closely match the percentage of a standard alloy. Aluminium 2024 is mainly alloyed with copper about 4 to 4.9 wt% which provides strength and promotes precipitation hardening. This type of alloy is usually complex due to presence of various alloying elements that match the property requirements such as corrosion resistance, strength, or grain structure control with copper, manganese, magnesium, and smaller amounts of silicon, chromium, zinc, iron and titanium. It has a muti-phase structure consisting of (Mn, Fe) 2SiAl1_2_, Mg_2_Si, CuAl_2_ and Al_2_CuMg [14,15,16].

Aluminium 6061 is a medium-to-high strength alloy with heat-treatable properties; the alloy has good corrosion resistance and manufacturability, as magnesium 1.16 wt% and silicon 1.51 wt% are the major alloying elements present in 6061. The inclusion of silicon (forms Mg_2_Si) improves resistance to abrasive wear and magnesium-silicon combination enhances strength of the alloy by heat treatments. Silicon also increases the followability of these alloys when casted [15]. Cr, Mn, Fe, and Zn trace elements are also present in the 6061 composition which contributes to its mechanical properties. The composition of Aluminium 7075 from the EDS majorly contains Zinc, Magnesium in moderate. It is composed of 5.54 wt% of zinc, 2.38 wt% of Magnesium and 1.6 wt% of Copper, Zinc in combination with magnesium, allow hardening heat treatment by precipitation, the presence of copper in 7075 increases its susceptibility to corrosion, but to make such a strong-yet-workable material, this trade is beneficial and is required. 7075 possesses excellent strength when compared to 6061 and 2024. Graphene was also studied with EDS in FE–SEM, the instrument records the output as carbon since graphene is an allotrope of carbon shown in Figure 9d.

### 3.5. FE-SEM/EDS Analysis of Microwave Processed and Hot Extruded Nano Composites

The fabricated composites that are studied for an alternative SLWT external tank structural application were prepared to observe in FE-SEM and analysed to gather reliable information on the elements present. On the selected area, EDS is performed on the composites under FE-SEM, as shown in Figure 10. The observations clearly exhibit effects of alloying elements forming stable intermetallic precipitates and dendrites; the presence of graphene is evident in the matrix from EDS and is read as carbon. The FE-SEM also reveals graphene agglomeration [15] at >2% weight ratio composites and poor weight proportion of the reinforcements in 1% Gr. Agglomeration of graphene will have adverse effects on the mechanical properties and other property aspects that do not match the existing SLWT external tank structural components properties; similar results are also reported in [16,17,18]. Graphene characterisation is difficult, as no clear evidence of the nature of graphene reinforced and changes in the nature of graphene can be found in this method. Additionally, nucleation and development of potential compounds such as Al_2_O_3_ and Al_4_C_3_ that have been formed cannot be identified using EDS. However, the introduced reinforcements were homogenously mixed and dispersed in the matrix of microwave processed and hot extruded composites.

### 3.6. XRD Analysis

The microwave-processed and hot-extruded composites that were studied for an alternative SLWT external tank structural application were studied for the compounds formed during and post to the fabrication process by using an X-ray diffraction method to identify the elements and potential crystalline structures formed from alloying elements and reinforcements in Aluminium alloys and composites. XRD pattern reads the crystal structure of the material based on its lattice parameters. Figure 11 are the XRD results obtained directly from the unit, the peaks for corresponding element can be read. In Figure 11 it is noted that carbon having a hexagonal crystal structure which can be correlated with hexagonal graphene (2024-Gr and 6061-Gr composites), whereas 7075-Gr composites graphene has been recorded as rhombohedral [19] crystal structure. This may be due to stack up of monolayers of graphene one over the other. Irrespective of alloy matrix used surface oxidation on the composite surface is noticed after the hot extrusion process as seen from Figure 11. Additionally, the nature of graphene is observed to be retained with its natural flakes forming after the microwave processing technique, and this mostly is a rapid process with involvement of higher activation energy at the grain interfaces and precipitate interfaces exhibiting low intensity of the formed Al_4_C_3_ compounds, which is highly favourable for the SLWT External tank structural application [20].

### 3.7. FTIR of Al-Gr Composites

Fourier Transform Infrared (FTIR) Spectroscopy is widely used as a method of elemental analysis for a wide range of inorganic materials, organic, polymeric, and two or more chemical compounds. In many studies, it is speculated that graphene present as reinforcements reacts with aluminium, forming aluminium carbides. This reaction may cause the composites to lose reinforcing properties, can degrade the processed composites, and is quite risky to use as a replacement composition for the SLWT external fuel tank structural application. However, Al_4_C_3_ has a tendency to react with moisture to hydrolyse and produce methane and is stable until 1400 °C [21]. Figure 12a–c shows a thin group from 820 cm^−1^ to 925 cm^−1^ (α) which can be associated with Al–O–Al bonds of anhydrous Al_2_O_3_, as seen in similar Al_4_C_3_ occurs at 499, 609, 711, and 785 cm^−1^, possibly due to Al–C bonding vibrations. Figure 11 show that most of the graphene particles remain as a reinforcing material without forming Al_4_C_3_. The possible explanation could be attributed to the maximum carbon solubility in aluminium being 0.015 wt% at PPM levels. This clearly proves that there is very low levels of Al_4_C_3_ formation in the composite matrix. Additional examination of FTIR Spectra from Figure 12a–c, exhibited peaks (φ) at ~1250 cm^−1^ and 1405 cm^−1^. Corresponding peaks are majorly due to the association of the C–O–C stretching with respective wavenumber and are mainly due to O–H deformation as a result of in-plane bending vibrations [22] research findings [22]. The standard FTIR absorption peaks for [19].

The stretching vibrations of sp_2_ C=C bonds peak, which is specific to the asymmetric flaky carbon in the form of graphene nano particles (GNPs) appearing at 1670 cm^−1^ in the spectrum recorded for the AA 7075 2 wt% Gr composite (β) [22]. Yet, a similar peak appeared in AA 6061 and AA 2024 2 wt% Gr at 1680 cm^−1^ with a stronger intensity that exhibited the form of exfoliated GNPs. This characteristics phenomenon can be related with flaky graphene carbon atoms interact with the incident ray, more widely and potentially owing to the exfoliation process as a result of ultra-sonification and microwave processing followed by hot extrusion of the composite. Reinforced nano GNPs exhibited no reaction and conversion of any intermetallic compound’s groups i.e., like Al_4_C_3_. It can also be understood that absorption bands in such phenomenon may be due to the detection boundary of the FTIR process for the solid samples [22,23,24].

## 4. Discussion

### 4.1. Electron Backscatter Diffraction (EBSD) Analysis

The EBSD maps and data show vital information such as average grain size diameter, grain elongation and misorientation angle, and pole figures, grains boundaries, and crystal structure orientation. Figure 13a–c shows the EBSD maps of the of AA 2024, AA 606, and AA 7075 with 2 wt% Gr composite in as microwave sintered condition, irrespective of the matrix used the graphene reinforcement facilitated in achieving ultra-fine grain boundary and also contributed well in enhancing the composite properties. Figure 13a–c confirms the ultra-fine-grained structure and found higher grain orientation along mixed (101), (111), and (001) composite microstructure, as most of the fabricated composite used average grain size of 10–25 µm (Figure 8). The addition of nano graphene flakes and its blend with microwave processing at the interface is confirmed using High resolution transmission electron microscope (HRTEM) and all the elemental maps (Figure 13a–c). This behaviour confirms that the excellent interface bonding between aluminium and nano graphene are achieved by microwave processing followed by hot extrusion process which is again confirmed and evident from HR-TEM analysis [25,26].

Comparing Figure 13 and Figure 14, the hot extrusion process induced a slightly elongated grain boundary, which leads to closure of micropores and other surface defects after the microwave sintering process. However, the presence of graphene evidently indicates the microstructure of the composites has a refined grain structure observed from the EBSD Maps and the pole figure from Figure 15 [20,21,22]. This can be attributed to the fact that the reinforcement added act as inhibitors of g-rain growth (pinning effect). The graphene’s high specific surface area (SSA) resulted in finer grain formation, number of atoms in contact with nucleation agent volume is increased with graphene and can be given by the equation.
(2)$$N_{het}=f_{1}C_{1}e^{\frac{\Delta G_{het}^{*}}{kT}}Nuclei/m^{3}$$
where *r*—radius of the particle, *f*—volume fraction (dispersed phase).

The refined grains, however, rely on the time profile and temperature, and depend on the critical size of the particle. These extruded samples produce elongated grains along with Graphene, during extrusion in the direction of metal flow, however larger weight fraction of graphene for above 2 wt% shows agglomeration of particles in the EBSD maps. These agglomerations make the grain boundaries weaker. The dispersed graphene particles are more clearly visible in higher magnification which can be ascertained from TEM analysis. It is also noted that the graphene remains as heterogeneous particle by not reacting with the phases and alloying elements present in the microstructure, the same can be verified in FTIR spectroscopy and TEM.

### 4.2. HR-TEM Analysis of Composites

Figure 16, Figure 17 and Figure 18 show graphene embedded in the AA2024, 6061 and 7075 matrixes, respectively. Composite is integrated with Nano-layered graphene (single atomic thickness), which reduces the spacing of the metal layer contributing to enhance the strength. 

The introduced graphene reinforcements mainly interrupt the growth of grains due to re-nucleation by graphene creating more fine grain structure. This is obvious, as the pinning effect of added graphene blocks the composite growth. Graphene acts a nano-level fillers thereby minimizing defects in the matrix material such as point defects, line dislocation, and surface defects. It was also noted that the interface between the matrix and graphene is effectively interlocked [25]. 

The pictures also displays precipitates found in the alloy matrix, which are responsible for improving material strength. These precipitates resist the movement and rearrangement of dislocations by means of pinning force. The optimum pinning power of dislocation is achieved over graphene, which stabilizes the microstructure by trapping the mobile dislocations. Two major challenges in the fabrication of Al–Gr composites are the formation of aluminium carbides with graphene and surface contamination of aluminium powders forming oxides. Few papers have reported the formation of Al_4_C_3_, whereas few others testified there is no transformation of Al_4_C_3_. 

This study also confirms there is no formation of carbides in the composite matrix. The possibility of carbide formation can be due one of several reasons, including (i) maximum solubility of carbon in aluminium [18], (ii) susceptibility of alloying elements present the matrix material [21,27] and (iii) reaction kinetics of Aluminium-carbon at elevated temperatures [19]. The maximum solubility of carbon in aluminium is reported to be 0.015 wt%. Moreover, the sintering temperature during fabrication was maintained at 450 °C, and these could be possible reasons for obtaining carbide-free composites.

Mitigation of oxide formation was yet another challenging task and can be addressed by fabricating the composite mixture in a vacuum/inert [20] atmospheric glove box and pre-heating the powders to remove any organic and hydrogen contaminants. Ball milling is also reported to break this oxide film [12,28]. However, the formation of oxides and carbides can be in the order of negligible wt% considering all these precautions.

### 4.3. Tensile Studies

The comparative findings obtained from the tensile test are compiled in Figure 19; the maximum strength exists at aluminium alloys reinforced by 2 wt% graphene. AA 7075 is also reported to have outperformed other alloys due to its inherent alloy properties. The maximum tensile strength observed for AA2024 + 2 wt% Gr, AA6061 + 2 wt% Gr and AA7075 + 2 wt% Gr is 255.5 MPa (parent alloy YS 97 MPa), 252.0 MPa (parent alloy YS 83 MPa), and 332.5 (parent alloy YS 140 MPa), respectively. Figure 20 shows the trend of proof stress and ductility of AA + 2 wt% graphene alloys; it is noted that alloys with 2% graphene of AA2024 showed high ductily in the order of 20.83%, AA6061 with 16.92% and the lowest for AA7075 with 12.28%. Tensile analysis obviously demonstrates increased tensile strength of aluminium alloys when properly reinforced with graphene nano-flakes; the strength improvement is substantial in the range of 25–30%. Improvements in the UTS due to the addition of graphene are also in good agreement with the work performed for aluminium by other scientists [17].

The increase in tensile strength demonstrates the effect of the addition of graphene in a virgin alloy as a suitable and promising reinforcement material. Increasing the mechanical properties meanwhile reflects the graphene’s elastic nature. Such a major UTS enhancement with the advent of graphene cannot simply be assumed to be a simple rule of mixtures, further its influence is far beyond what was previously reported and can be extensively studied [29,30,31,32,33]. The presence of the graphene flakes in the composite has a strong influence on composite material design criteria.

The strengthening is due to load transferring from the matrix material to graphene particle reinforcements, which also act as a load bearing component and not merely controlling dislocation movement. There is a small drop in strength due to agglomeration in the higher concentrations of graphene 2.5% and 3%, and lower concentrations are not adequate to bear matrix loads. The stress–strain curves generated during the experiments are shown in Figure 21; the curve is linear until plastic deformation occurs, and proof stress values can be obtained from the same. It can be assumed that fine particles of graphene resemble precipitation hardening system in the matrix material forming fine clusters, and this clustering may produce local strain which is transferred effectively to the reinforcements [22,34,35].

The degree of strengthening depends on the particle distribution in the ductile matrix; in addition to shape of the particles, the reinforcements can be described by specifying average particle size, volume fraction, and mean interparticle spacing. Graphene prevents the propagation of cracks thereby strengthening the matrix by absorbing the dislocations from the matrix material. Such reinforcements work in two distinct ways to delay dislocation movement: the reinforcement particles can be cut by dislocations, or the graphene particles can resist dislocation [34].

It is also known that graphene’s planar 2-D structure interlocks the aluminium matrix thus establishing a diffusion-free bond between both increasing the dislocation density by thermal disparity between filler and the matrix. Such 2-D planar structures can be in the form folds and wrinkles, when the material is loaded the load transfer to reinforcements either try to unfold or expand the graphene particles. The other predictable reason for strength increase is due to grain fineness by graphene particles due to thermal heat absorption, the grain fineness can be examined in a microstructural investigation.

Hall-Petch [10] grain refinement strengthening can be described to attain the strength in the composite given in Equation (3) [32,36,37,38,39]
σ_y∝√d(3)
where, d—average diameter of the grain size.

The other strength mechanisms could be by inherent properties of the alloy attributed to stacking fault energy, coherency strain, lattice friction stress, modulus effect, ordered structure, and interfacial energy.

The strain field resulting from the discrepancy between particle and the matrix given by Mott and Nabarro would be a basis of strengthening. The increase in yield strength is given by Equation (4) [27]
∆σ = 2 (G × ε × f)(4)
where f—volume fraction (dispersed phase) and ε—measure of the strain field.

The energy of dislocations also depends linearly on the local modulus, particles which have a modulus which differs significantly from the matrix material will raise or lower the energy of a dislocation as it passes through them. This strengthening effect is given by Equation (5)
∆σ = (∆G/2π)^2^ [3ǀ∆Gǀ/Gb]^(1/2)^ [0.8 − 0.143 ln
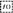

(r/b)]^(3/2)^ r^(1/2)^ f^(1/2)^(5)
where f is the volume fraction (dispersed phase) and r is the radius of the particle.

In most alloys, there is not enough of a modulus difference; however, the reinforcement fulfils the criteria for the difference in modulus, thereby producing a possible strong strengthening effect. The above equation given by Hirsch and Kelly is derived for the case of spherical reinforcements, but the maximum strengthening due to surface effects arises from thin plate-shaped (planar) reinforcements which can resemble graphene’s planar structure.

The significant improvement in the ultimate tensile strength, results in a longer life with nominal ductility. These enhancements of composite strength are attributed mainly to strengthening of dislocation, optimum grain refinement, and various other models. The Orowan strengthening mechanism may also contribute to improvement in the strength of the composites. It can also be justified that the extrusion process is also leading to an improvement in tensile strength; however, it is challenging to understand the contributions from two or more mechanisms combined [22].

### 4.4. Fracture Morpology

Figure 22, Figure 23 and Figure 24 show dimpled rupture characterised by cup-like depressions that are almost equiaxial in nature and reflects the SEM images of the fracture surface, showing ductile composite behaviour due to micro-void coalescence in the molecular level of mixing. The ductile fracture characteristics are observed for all grades containing 2 wt% Gr. This can be attributed to uniform distribution of graphene inside the grains providing rigid interfacial areas; these interfacial areas act as both dislocation initiation sites and dislocation sinks pinning the dislocation motion at the same time. The uniform distribution resisted accumulation of reinforcements at the grain boundaries in 2 wt% Gr composites, allowing good strength to be demonstrated by dislocation pinning without losing the composite ductility. Anything less than 2 wt% Gr shows scattered and dispersed reinforcements exposing more of matrix material, which articulates to more of matrix properties. While greater than 2%, i.e., 2.5% and 3% of graphene reinforcements are represented by dimples and tear ridges known as quasi-cleavage fracture due to the agglomeration in the grain boundaries rendering weak tensile strength due to weak interface bonding.

Figure 22a–c presents the Fractography images AA 2024 grade with varying percentage of graphene at 50 KX magnifications, the graphene distribution is clearly visible for all the reinforcement percentage, the inherent property of alloying elements forming multiphase structure also contributed to increase in strength of the composite. The Fractography images of AA 6061 grade with graphene is shown from Figure 23a–c. It reveals that added graphene were aligned randomly in all directions.

The graphene pulls and tear signifies that the transition of load occurred from the matrix to the reinforcement, and partially pulled out and pulled off graphene was also observed which confirms the transformation of load from matrix to graphene. Figure 24a–c shows 50 kX magnification images of AA 7075 reinforced with 1.5%, 2% and 3% of graphed displays finely dispersed reinforcement at 2% and dimples are shallow exhibiting good ductility without losing strength.

### 4.5. Hardness Evaluation

Figure 25 shows the combined hardness of different grade Aluminium samples reinforced with graphene. It is clear from the data that AA 7075 shows the highest hardness values with 2 wt% graphene, and the lowest is AA 6061 with 1 wt% graphene. Since the tests of the tensile strength are appropriate for composites reinforced with 2 wt% graphene, it would be suitable to discus and describe the 2 wt% reinforced graphene for further analysis. Compared to the basic standard alloy hardness values, there is also an overall increase in hardness for graphene-reinforced Aluminium [40]. This increase in hardness may basically be associated with the grain fineness in the composites observed in the microstructure analysis. Graphene acts as a grain refiner due to differences in thermal interface properties and also acts as a void filler, which can further inhibit the dislocation movement, improving the material’s strength and durability. The increasing hardness in correlation with tensile strength is due to the fact that composites were subjected to strain hardening during the extrusion process. The effect of strain hardening also improved the morphology for reinforcement and the matrix bonding. The hardness is maximum 95 HRB for 7075 + 2% Gr and the lowest 63 HRB is recorded for AA 6061 + 1% Gr.

## 5. Conclusions

In this research study, an exhaustive characterisation such as spectroscopy, XRD, EDS, EBSD, FE-SEM, and HR-TEM analysis was carried out to determine the metallurgical properties, chemical analysis, and formation of detrimental phases, and mechanical properties such as tensile and hardness were investigated. The findings from the characterisation and mechanical evaluation were compared potentially with the materials requirements of the SLWT external fuel tank structural components, indicating the following clarifications.

Graphene nano-flake analysis at higher magnification revealed that graphene is a two–dimensional flaky structure. It was also noted that graphene has an enormous surface area, which plays a key role as matrix material reinforcements with good wettability and can be a potential reinforcing material when considered for the launch vehicle SLWT external tank structural application.

The composites were successfully fabricated and the reinforcement particles were found to be homogenously distributed and clarified through EBSD maps and HR-TEM. EBSD of 2 wt% Gr shows more uniform dispersion of particles; however, higher wt% Gr exhibited grain boundary agglomeration. Microwave processing and hot extrusion (comparable to rolled AA 2195 Airware [14,41]) processing conditions served the composite to attain its potential to be considered for the existing material for the SLWT external tank structural application.

Graphene remains as a heterogeneous particle by not reacting with the phases and alloying elements present in the microstructure, the same can be also verified in FTIR spectroscopy, and TEM.XRD, EDS, and FTIR confirm the presence of graphene was intact as a reinforcement material and that there was no formation of carbide. This non-reactive nature of the graphene is achieved by the microwave processing, as the process is quite fast and heat is achieved from the core of the composite.

The maximum yield strength exists at aluminium alloys reinforced by 2 wt% graphene. AA 7075 is also reported to have outperformed other alloys due to its inherent alloy properties. The maximum tensile yield strength observed for AA2024 + 2% Gr, AA6061 + 2% Gr, and AA7075 + 2% Gr was 255.5 MPa, 252.0 MPa, and 332.5, respectively. AA 7075 with 2% graphene displayed the highest hardness values, and the lowest was AA 6061, with 1% graphene. The hardness was a maximum of 95 HRB for 7075 + 2% Gr and the lowest 63 HRB was reported for AA 6061 + 1 wt% Gr.

The graphene pulls and tears signify that the transition of load occurred from the matrix to the reinforcement, and partially pulled out and pulled off graphene was also observed, which confirms the transformation of load from matrix to graphene. The bonding mechanism, processing conditions offered by microwave sintering, and the hot extrusion process (compared to hot rolling process) were found to be potential and perfect conditions to convert the fabricated composites for the launch vehicle SLWT external tank structural application.

## Figures and Tables

**Figure 1 materials-15-05907-f001:**
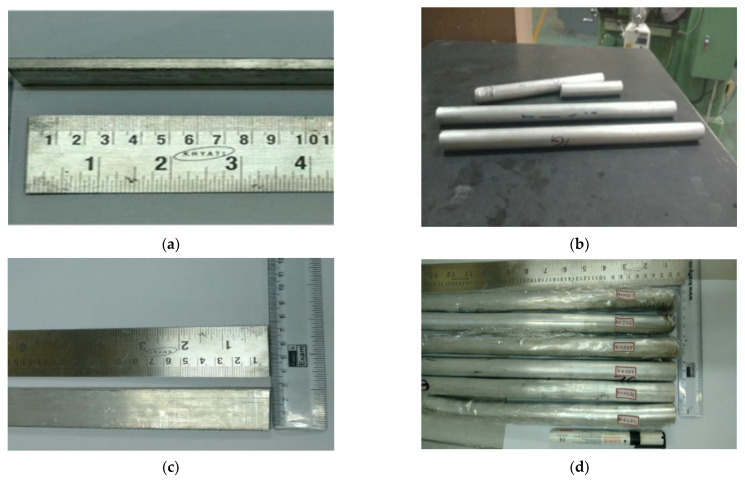
Extruded composites with 2 wt% graphene; (**a**,**c**) are the flat bar, and (**b**,**d**) are the round bar.

**Figure 2 materials-15-05907-f002:**
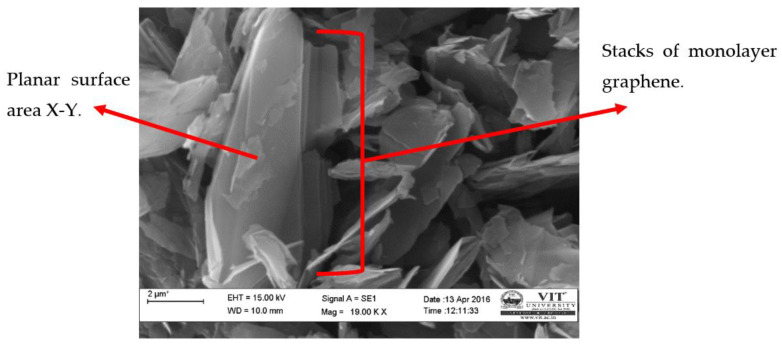
FE-SEM image of as received graphene Nano flakes.

**Figure 3 materials-15-05907-f003:**
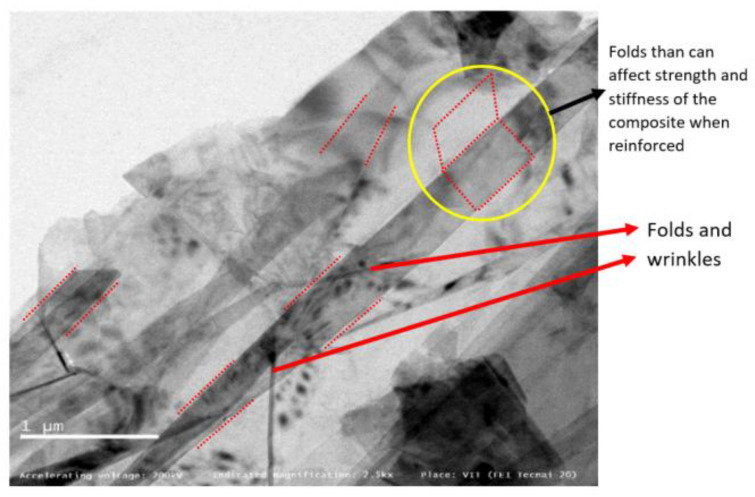
HR-TEM observation of graphene at 2500×.

**Figure 4 materials-15-05907-f004:**
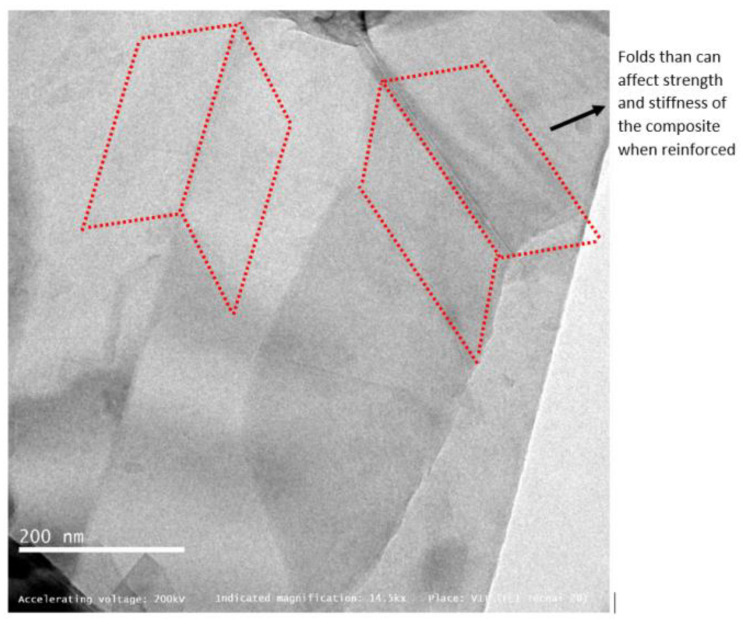
HR-TEM observation of graphene at 14,500×.

**Figure 5 materials-15-05907-f005:**
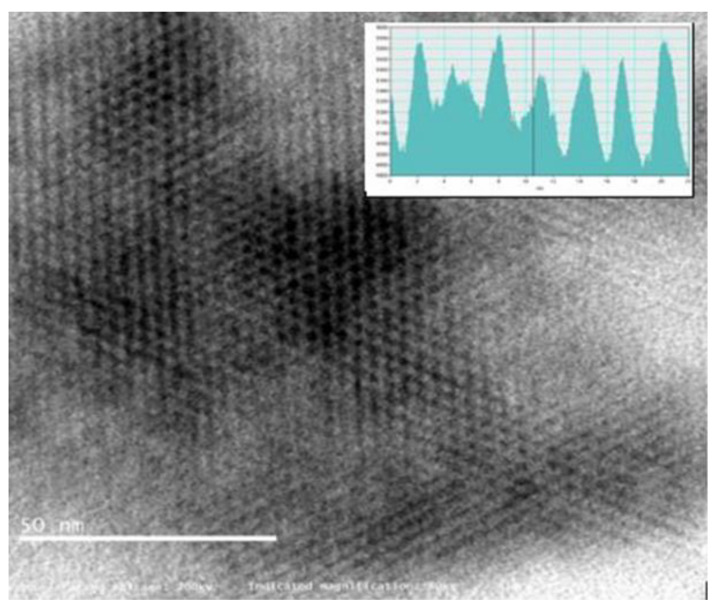
HR-TEM observation of graphene at 80,000×.

**Figure 6 materials-15-05907-f006:**
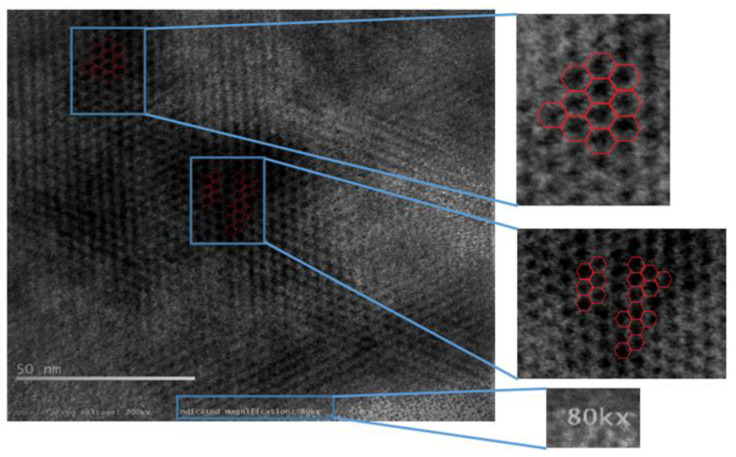
Enhanced and processed image of graphene at 80,000× from HR-TEM.

**Figure 7 materials-15-05907-f007:**
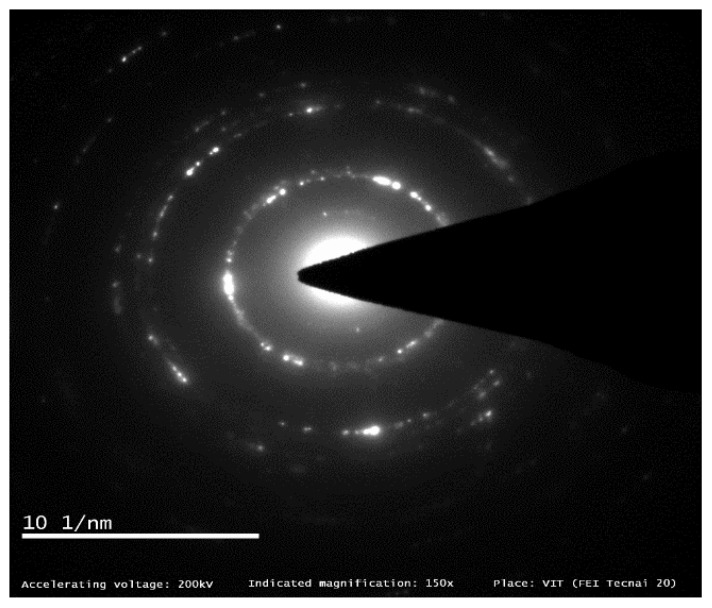
HR-TEM. Electron Diffraction pattern of graphene obtained from HR-TEM.

**Figure 8 materials-15-05907-f008:**
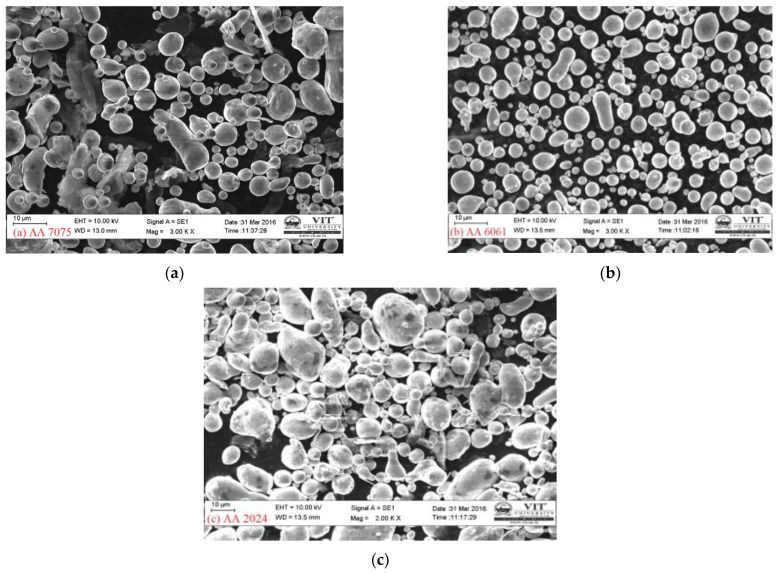
FESEM pictures of received (**a**) AA 7075, (**b**) AA 6061, (**c**) AA 2024.

**Figure 9 materials-15-05907-f009:**
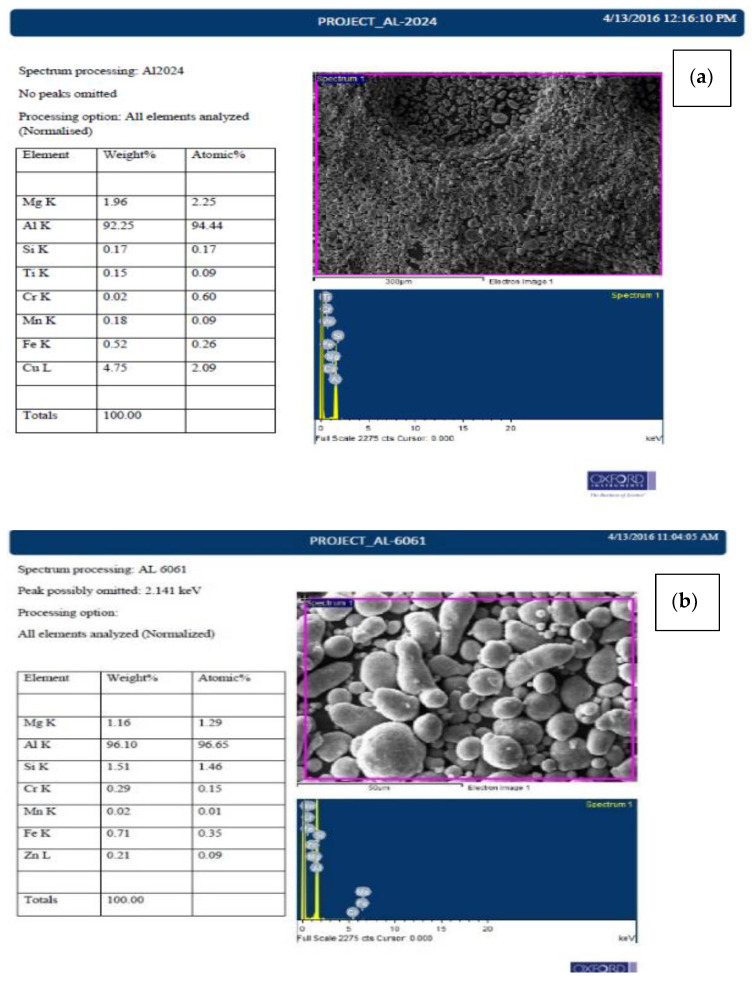
FESEM pictures of received (**a**) AA 2024, (**b**) AA 6061, (**c**) AA 7075 and (**d**) Graphene.

**Figure 10 materials-15-05907-f010:**
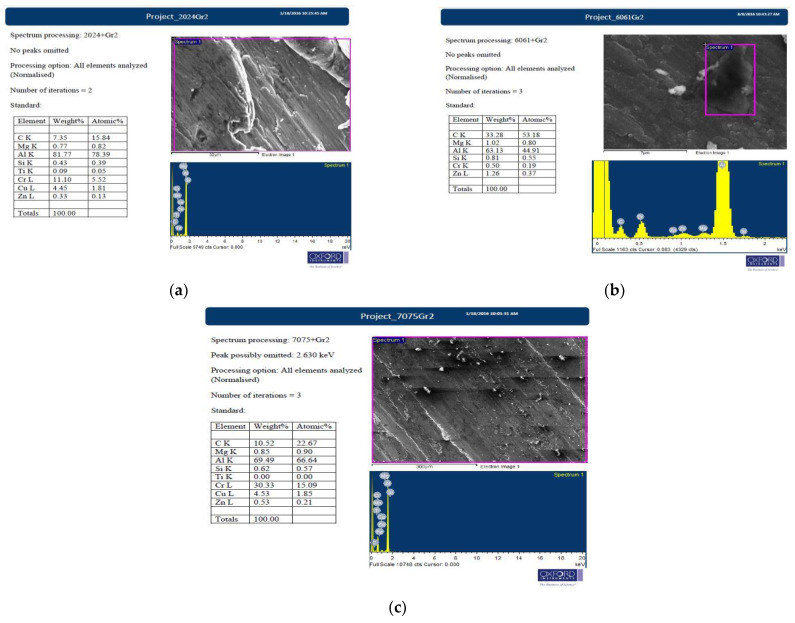
EDS analyzed elemental composition of fabricated composites (**a**) AA2024 + 2%Gr, (**b**) AA6061 + 2%Gr and (**c**) AA7075 + 2%Gr.

**Figure 11 materials-15-05907-f011:**
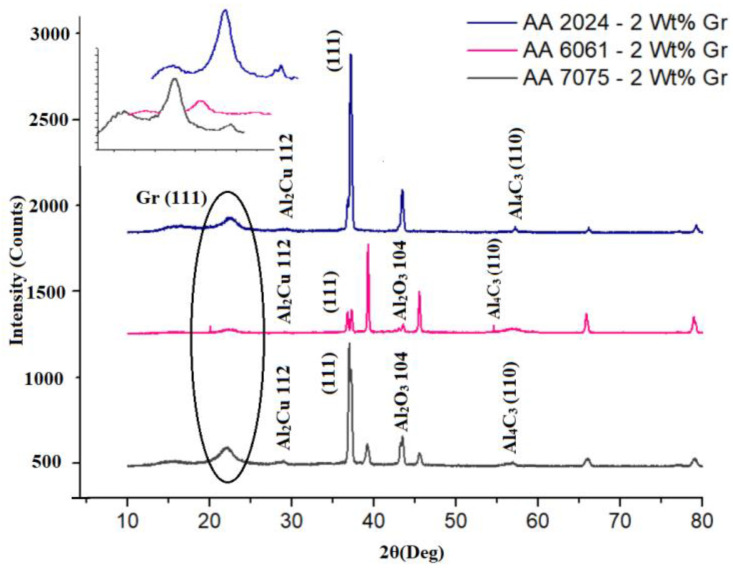
XRD Pattern for AA2024, AA6061 and AA7075–2 Wt% Gr Nano Composites.

**Figure 12 materials-15-05907-f012:**
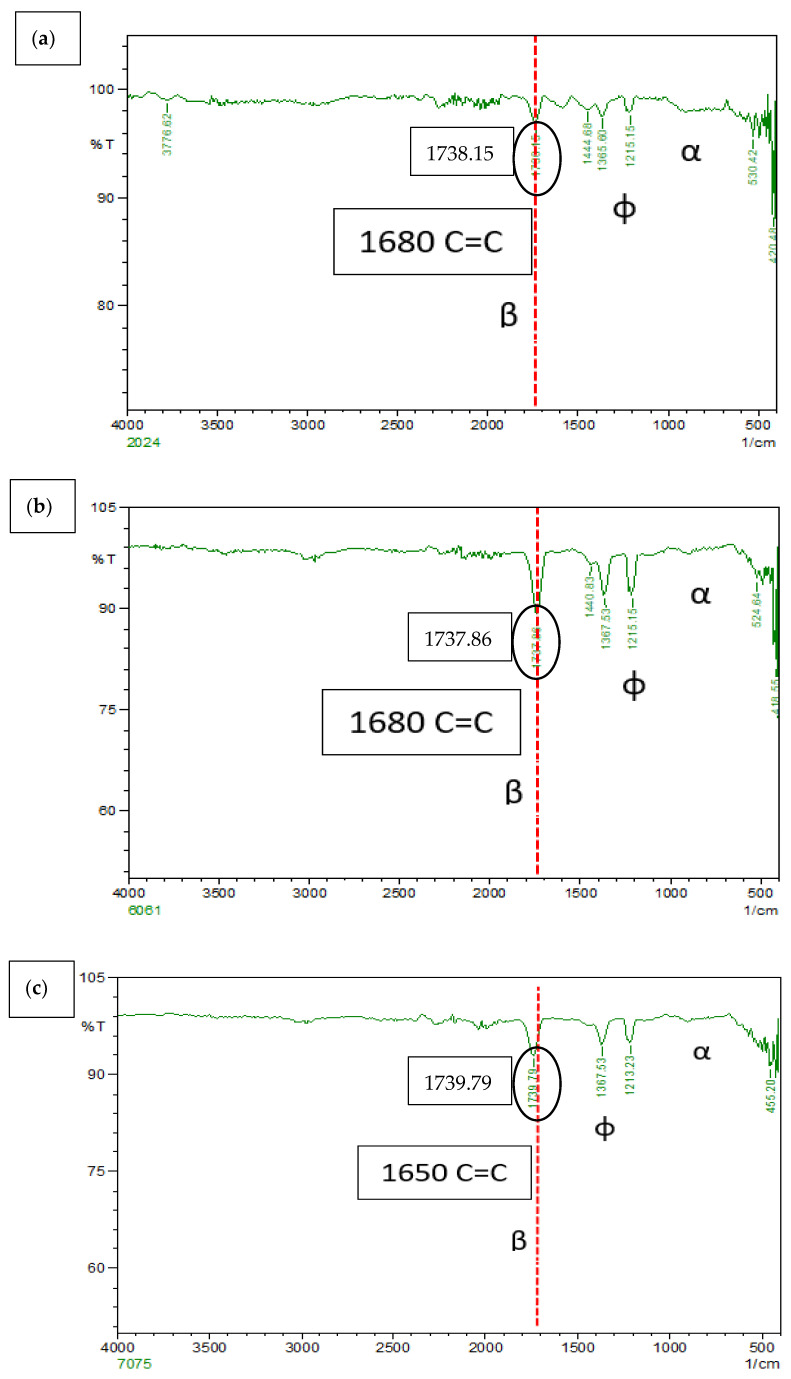
FTIR peaks for Microwave processed and hot extruded (**a**) 2024 + Gr, (**b**) 6061 + Gr and (**c**) 7075 + Gr.

**Figure 13 materials-15-05907-f013:**
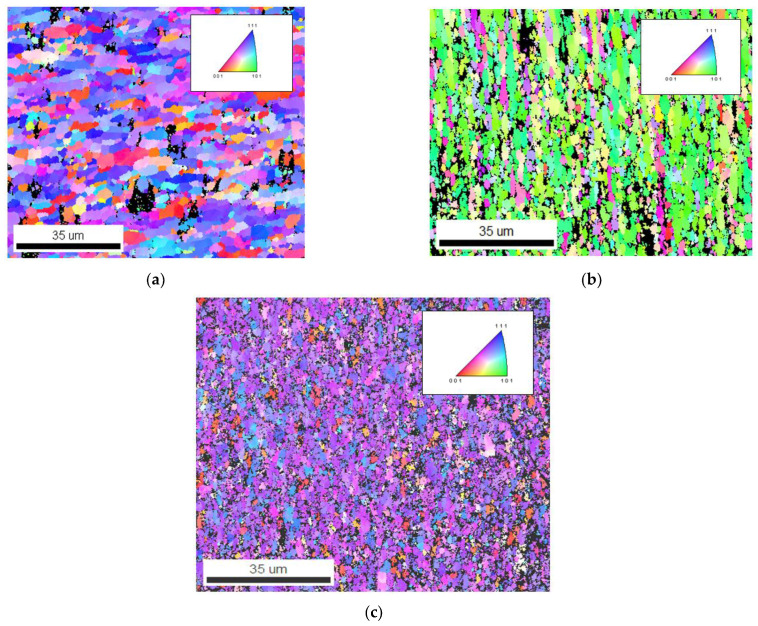
EBSD Maps of as sintered composite billets (**a**) AA 2024–2 wt% Graphene composites (**b**) AA 6061–2 wt% Graphene composites (**c**) AA 7075–2 wt% Graphene composite.

**Figure 14 materials-15-05907-f014:**
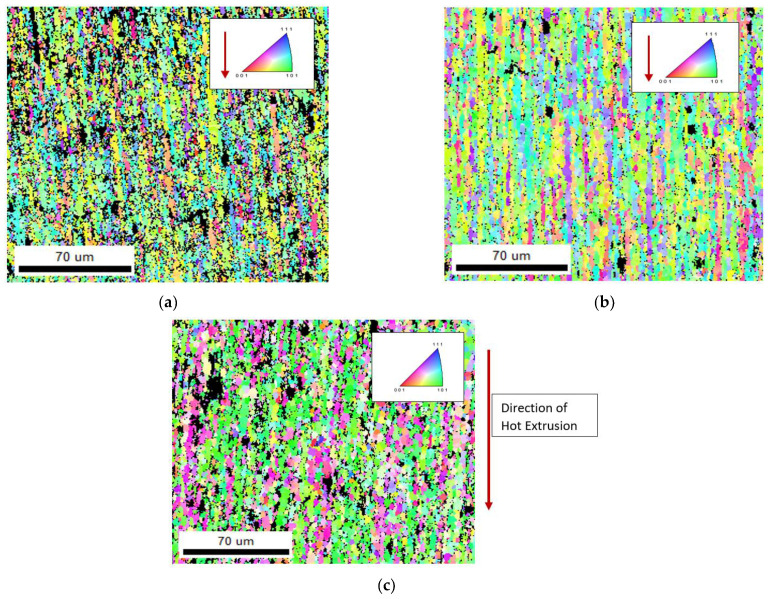
EBSD Maps of Hot extruded composite plates (**a**) AA 2024–2 wt% Graphene composites (**b**) AA 6061–2 wt% Graphene composites (**c**) AA 7075–2 wt% Graphene composite.

**Figure 15 materials-15-05907-f015:**
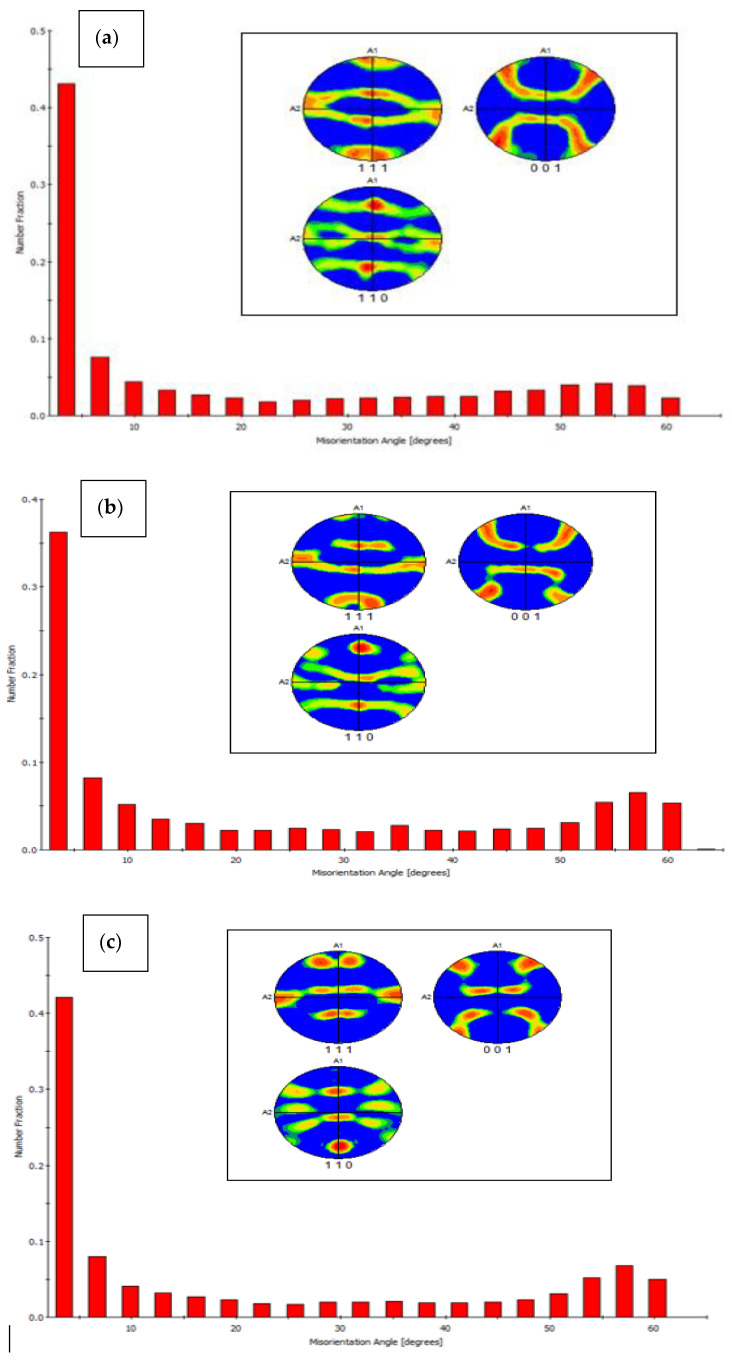
EBSD grain misorientation angle and pole figure of Hot extruded composite plates (**a**) AA 2024–2 wt% Graphene composites (**b**) AA 6061–2 wt% Graphene composites (**c**) AA 7075–2 wt% Graphene composite.

**Figure 16 materials-15-05907-f016:**
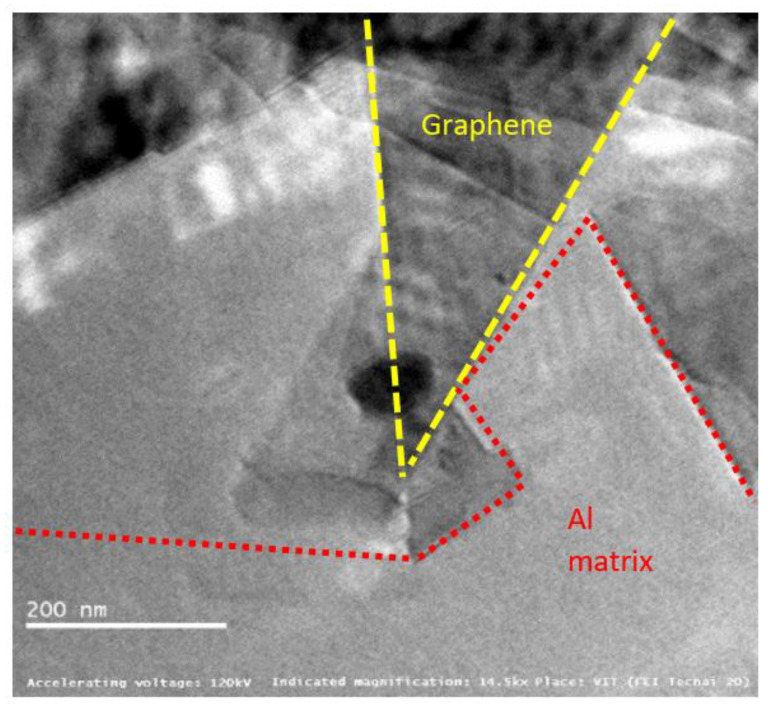
Graphene sheets embedded in AA2024 at 14.5 kX.

**Figure 17 materials-15-05907-f017:**
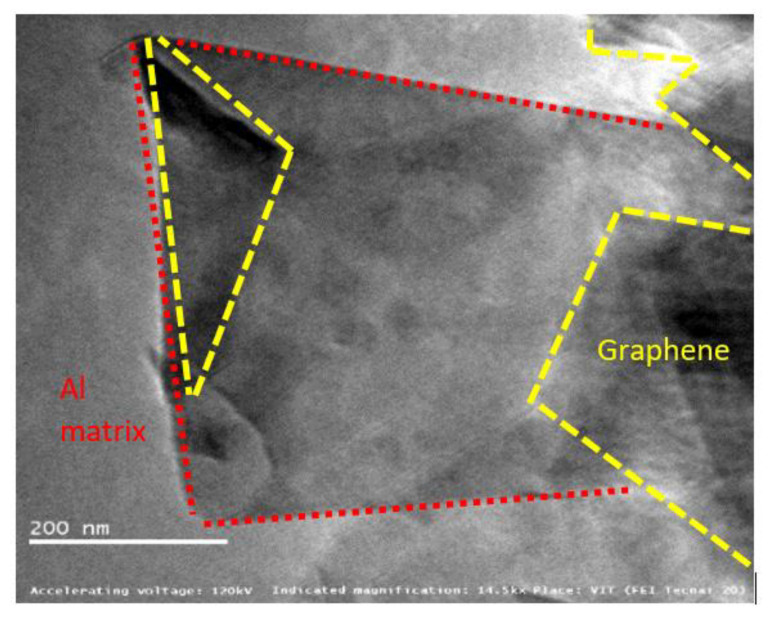
Graphene sheets embedded in AA6061 at 14.5 kX.

**Figure 18 materials-15-05907-f018:**
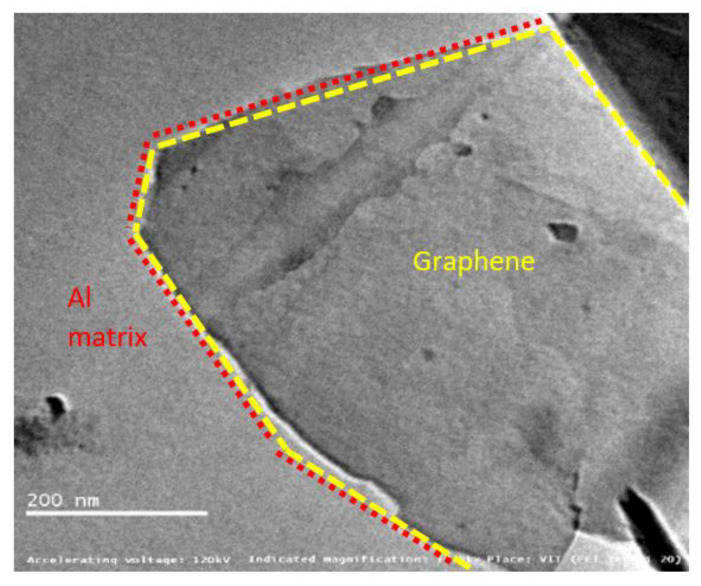
Graphene sheets embedded in AA7075 at 14.5 kX.

**Figure 19 materials-15-05907-f019:**
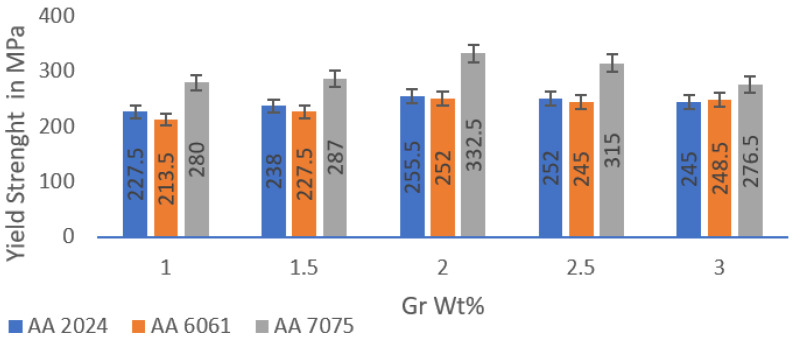
Yield Strength data of the microwave-processed Hot extruded Graphene Nano composites.

**Figure 20 materials-15-05907-f020:**
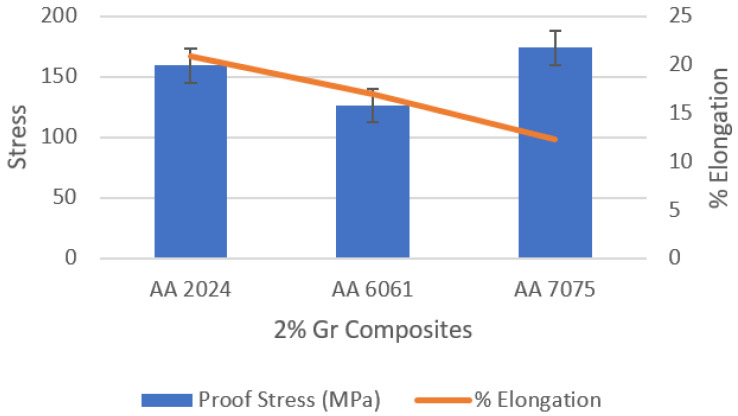
Trend of proof stress and elongation.

**Figure 21 materials-15-05907-f021:**
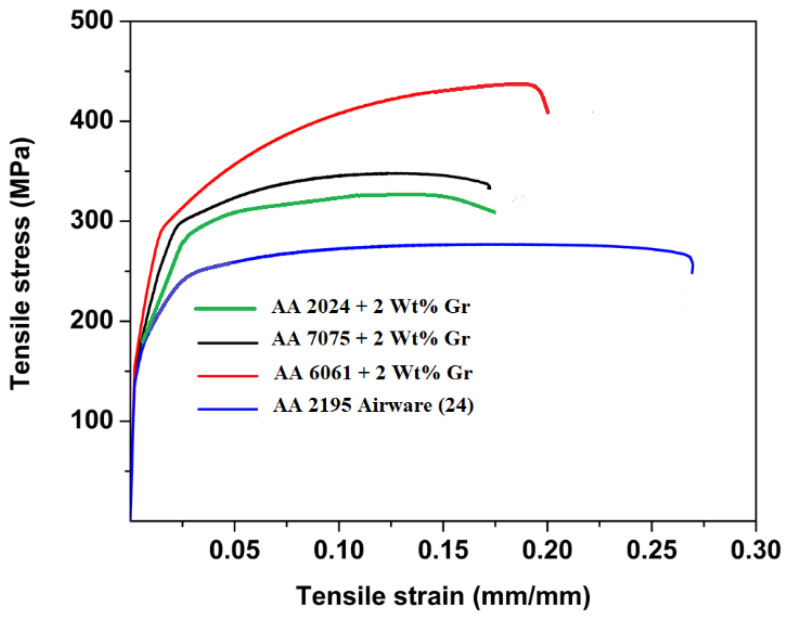
Stress–strain curve comparison between hot extruded graphene nano composites with Aluminium lithium alloys (non-heat treated) (24) used for launch vehicle SLWT external fuel tank structural applications.

**Figure 22 materials-15-05907-f022:**
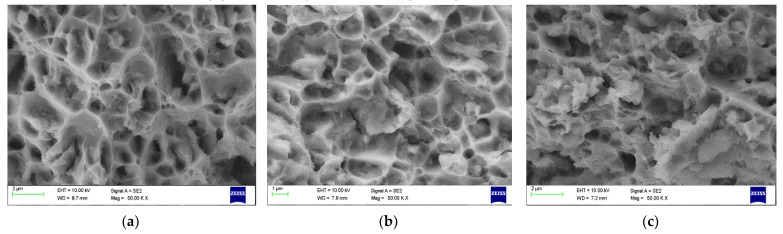
Fractography SEM image of 2024 (**a**) 1% Gr (**b**) 2%Gr (**c**) 3%Gr.

**Figure 23 materials-15-05907-f023:**
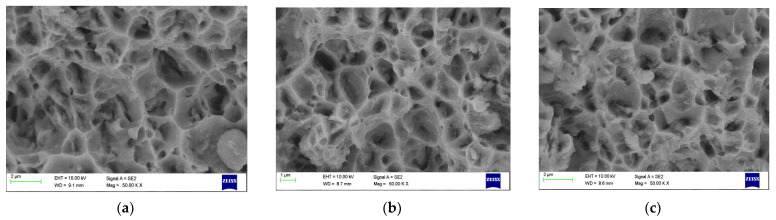
Fractography SEM image of 6061 (**a**) 1% Gr (**b**) 2%Gr (**c**) 3%Gr.

**Figure 24 materials-15-05907-f024:**
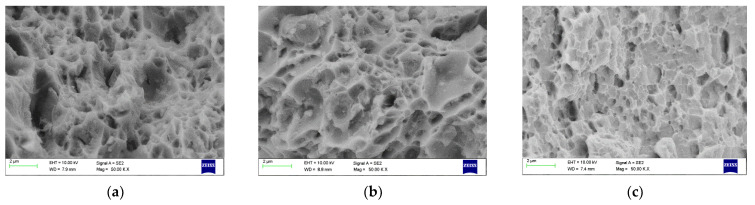
Fractography SEM image of 7075 (**a**) 1% Gr (**b**) 2%Gr (**c**) 3%Gr.

**Figure 25 materials-15-05907-f025:**
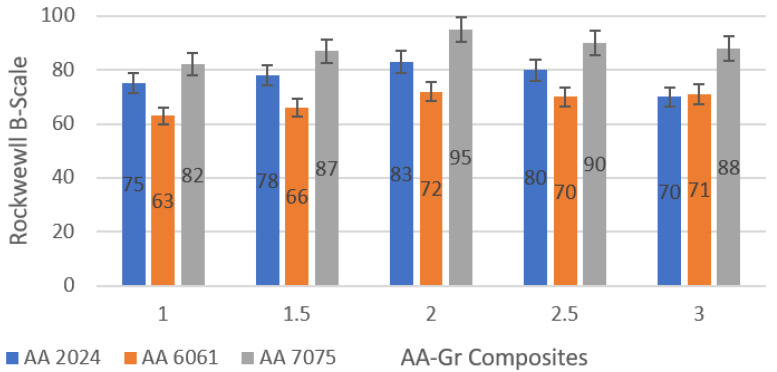
Hardness Evaluation of AA-Gr Composites.

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
