# Peer review of "Characterization Studies on Graphene-Aluminium Nano Composites for Aerospace Launch Vehicle External Fuel Tank Structural Application"

_materials, 2022, doi:10.3390/ma15175907_

Round 1
Reviewer 1 Report
The manuscript titled “Characterization studies on Graphene-Aluminium Nano composites for aerospace launch vehicle external fuel tank structural application” (Materials 1850923) is a comprehensive study on the effect of graphene-aluminum nano-composites. It is a comprehensive manuscript with a plethora of characterization techniques, that the readers of Materials would find interesting. The authors should address the following comments prior to publication consideration.
Main comment: Comment 1. Figure 11 (XRD) shows a small peak at around 2theta ~ 55 degrees labeled as Al4C3 (110) for all composites shown. This peak is associated with aluminum carbide content (see reference https://doi.org/10.1016/j.msea.2016.01.076 for example). However, the authors spend much of the subsequent discussion to reinforce there is no Al4C3 in the composites. For example, in lines 292-293 “This clearly shows that there is no any likelihood of Al4C3 formation in the composite matrix”, lines 370 “This study confirms there is no formation of carbides in the composite matrix”. Can the authors explain why when the XRD shows formation of aluminum carbide in the 2 wt% composites, they claim there is no aluminum carbide formation?
Comment 2. Tensile studies: How do these results compare to neat AA2024, AA6061, AA7075?
Comment 3. I believe the reference list inadequate, given the length and breadth of the manuscript. Please consider adding more references, including the following:
Afifah Md Ali, et al., Recent development in graphene-reinforced aluminium matrix composite: A review, Reviews on Advanced Materials Science 2021; 60: 801–817.
Stephen F. Bartolucci, et al., Graphene–aluminum nanocomposites, Materials Science and Engineering A 528 (2011) 7933– 7937.
Zhang et al., Enhanced mechanical properties of Al5083 alloy with graph-ene nanoplates prepared by ball milling and hot extrusion, Materials Science and Engineering: A 658, 8-15 (2016) //discussion of Al4C3 XRD peaks
Author Response
Response to First Reviewer
Reviewer#1:
- Figure 11 (XRD) shows a small peak at around 2theta ~ 55 degrees labeled as Al4C3 (110) for all composites shown. This peak is associated with aluminum carbide content (see reference https://doi.org/10.1016/j.msea.2016.01.076 for example). However, the authors spend much of the subsequent discussion to reinforce there is no Al4C3 in the composites.
For example, in lines 292-293 “This clearly shows that there is no any likelihood of Al4C3 formation in the composite matrix”, lines 370 “This study confirms there is no formation of carbides in the composite matrix”.
Can the authors explain why when the XRD shows formation of aluminum carbide in the 2 wt% composites, they claim there is no aluminum carbide formation?
Ans: From figure 11 it is evident that there were few traces of the Al4C3, however the authors intend to mention the negligible aspect of carbide formation. The technical discussion related to XRD results were modified which will give more clarity on the carbide formation when processing the composites. It is also included in the technical write up that 2 wt% composite also was exhibiting few traces of carbide formation, this phenomenon is majorly due to the carbon solubility in aluminium matrix that are being limited to 0.015 wt % at PPM levels.
- How do these results compare to neat AA2024, AA6061, AA7075?
Ans: In the tensile studies the yeild strength are compared to the parent materials yield strength which is clearly included in the subsection.
- I believe the reference list inadequate, given the length and breadth of the manuscript. Please consider adding more references, including the following:
Ans: The manuscript is updated with the recent literature and research findings that are highly relevent to the current research. Also, we have included the necessary references suggested by the reviewers.
- Minor corrections related to the manuscript grammar and language.
Ans: The manuscript is checked and reviewed by a native English speaker and all the necessary corrections were incorporated and updated in the current manuscript.

Reviewer 2 Report
Some relevant references for the topic of this paper must be included:
Venkatesan, S., & Xavior, M. A. (2019). Characterization on aluminum alloy 7050 metal matrix composite reinforced with graphene nanoparticles. Procedia manufacturing, 30, 120-127.
Venkatesan, S., & Xavior, M. A. (2018). Tensile behavior of aluminum alloy (AA7050) metal matrix composite reinforced with graphene fabricated by stir and squeeze cast processes. Science and technology of materials, 30(2), 74-85.
Singh, P. K. (2021). Mechanical characterization of graphene-aluminum nanocomposites. Materials Today: Proceedings, 44, 2304-2308.
Awate, P. P., & Barve, S. B. (2022). Enhanced microstructure and mechanical properties of Al6061 alloy via graphene nanoplates reinforcement fabricated by stir casting. Functional Composites and Structures, 4(1), 015005.
The authors must clearly emphasize the novelty of this paper.
The resolution of Fig 22, 23 must be improved.
Author Response
Response to Second Reviewer
- Some relevant references for the topic of this paper must be included:
Ans: The manuscript is updated with the recent literature and research findings that are highly relevant to the current research. Also, we have included the necessary references suggested by the reviewers.
- The authors must clearly emphasize the novelty of this paper.
Ans: In the abstract and through out the manuscript the novelty of the research is mentioned and maintained. As per the suggestion given by the reviewer, few technical aspects were added which further improves the quality of the manuscript.
- The resolution of Fig 22, 23 must be improved.
Ans: The resolution of figure 22 and 23 is improved as suggested.

Reviewer 3 Report
Title: Characterization studies on Graphene-Aluminum Nano composites for aerospace launch vehicle external fuel tank structural application
The referee would like to recommend this work to major revision and to be published after consideration according to the comments below:
1. Please add more current papers in the literature and improve introduction section. Some interesting papers related to the topic of this manuscript could be:
Experimental and numerical study on HDPE/SWCNT nanocomposite elastic properties considering the processing techniques effect. Microsystem Technologies, 2020, 26, 2423–2441.
Development of efficient size-dependent plate models for axial buckling of single-layered graphene nano sheets using molecular dynamics simulation. 2018, Microsystem Technologies, 24(2), 1265-1277.
2. Some Figures (For example: Figure 13, 14) and Equations( For example: Equation (5)) are illegible and unclear. Please correct them.
Author Response
Response to Third Reviewer
- Please add more current papers in the literature and improve introduction section. Some interesting papers related to the topic of this manuscript could be:
Ans: The manuscript is updated with the recent literature and research findings. Introduction section is further enhanced by adding relevant findings of the current research. Also, we have included the necessary references suggested by the reviewers.
- Some Figures (For example: Figure 13, 14) and Equations (For example: Equation (5)) are illegible and unclear. Please correct them.
Ans: All the figures appearing in the manuscript are checked and good quality figures are presented in the revised manuscript as per the quality standard mentioned by the journal and the equations are re written as suggested.

Round 2
Reviewer 1 Report
The authors have addressed my comments, therefore I recommend the manuscript for publication.